# ALMA: Hierarchical Learning
# for Composite Multi-Agent Tasks

**Shariq Iqbal**[*]
Deepmind

**Robby Costales**
University of Southern California

**Fei Sha**
Google Research

## Abstract

Despite significant progress on multi-agent reinforcement learning (MARL) in recent years, coordination in complex domains remains a challenge. Work in MARL often focuses on solving tasks where agents interact with *all* other agents and entities in the environment; however, we observe that real-world tasks are often *composed* of several isolated instances of local agent interactions (subtasks), and each agent can meaningfully focus on one subtask to the exclusion of all else in the environment. In these *composite tasks*, successful policies can often be decomposed into two levels of decision-making: agents are *allocated* to specific subtasks and each agent *acts* productively towards their assigned subtask alone. This decomposed decision making provides a strong structural inductive bias, significantly reduces agent observation spaces, and encourages subtask-specific policies to be reused and composed during training, as opposed to treating each new composition of subtasks as unique. We introduce ALMA, a general learning method for taking advantage of these structured tasks. ALMA simultaneously learns a high-level subtask allocation policy and low-level agent policies. We demonstrate that ALMA learns sophisticated coordination behavior in a number of challenging environments, outperforming strong baselines. ALMA's modularity also enables it to better generalize to new environment configurations. Finally, we find that while ALMA can integrate separately trained allocation and action policies, the best performance is obtained only by training all components jointly. Our code is available at https://github.com/shariqiqbal2810/ALMA

## 1 Introduction

A certain set of real-world multi-agent tasks increase in complexity in proportion to the inter-agent coordination required rather than any inherent difficulty in the low-level skills. Take the example of firefighting (Figure 1): while coordinating firefighters across a large city is more complex than in a small one, the individual sub-task of putting out a fire remains similar across cities of all sizes. Despite the recent success of cooperative multi-agent reinforcement learning (MARL) methods on a wide range of domains [2, 16, 35, 5], they still struggle to learn sophisticated coordination behavior in a sample-efficient manner on tasks of real-world complexity. And this is no surprise—not only do these tasks require that agents learn complex low-level skills to execute sub-tasks, they must also incorporate coherent high-level strategies into their policies. Integrating both strategies and skills into the same action-level policy makes learning difficult.

Fortunately, for many relevant multi-agent tasks, while there may exist many different objectives that the agent population *as a whole* must attend to, *individual* agents need only focus on isolated aspects of the environment at any given time. Consider again the setting of firefighting in a city given a distributed set of resources. While there may be many fires occurring in the city at once, each firefighter can realistically only fight one at a time. Thus, the optimal behavior in this setting can be

---

[*]Work performed while at USC.

36th Conference on Neural Information Processing Systems (NeurIPS 2022).

expressed in terms of allocating firefighters to the right fires, and each firefighter effectively fighting the fire to which they are assigned.

Suitable policies for this and many other real-world problems can be formulated as bi-level decision-making processes [20]; agents are allocated to the most relevant *subtasks* in the environment, then each agent selects actions with the purpose of completing its assigned subtask. We call settings that can be formulated in this way "composite tasks". Decomposing these tasks allows for more efficient learning for a number of reasons. First, the learning process may benefit from the added structural inductive bias, alleviating the problem of incorporating both skills and strategy into the same low-level policy. Second, by focusing solely on its assigned subtask, and ignoring the rest of the environment, agent action-level policies can learn over

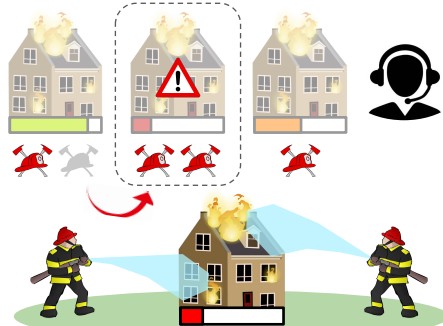

Figure 1: The task of city-wide firefighting can be decomposed into allocating firefighters to the most urgent fires (top), and firefighters focusing solely on the fire they are assigned to (bottom).

a significantly reduced state-space. Finally, as we assume subtasks are drawn from the same distribution, learning policies for individual subtasks and allowing the high-level controller to compose them may be simpler than learning a single policy over all compositions of these subtasks.

Our contributions are as follows. First, we introduce a general learning method, **AL**locator-Actor **M**ulti-**A**gent **A**rchitecture (**ALMA**), for taking advantage of composite multi-agent tasks (§4). **ALMA** learns a subtask allocation policy and low-level agent policies simultaneously and is designed in a modular manner to promote the reuse of components across different compositions of subtasks. Learning the high-level controller poses challenges due to its large action space—at any time step, the number of unique allocations of $n$ agents to $m$ subtasks is $m^n$. Our solution leverages recent methods [31] in learning value functions over massive action spaces and is designed modularly to handle variable quantities of agents and subtasks drawn from a shared distribution. While action-value function typically depend on the global state due to shared transition dynamics and reward functions that depend on *all* agents, we observe that for composite tasks we can decompose the environment into independent sub-environments and redefine subtask-specific action-value functions to only depend on *local* information, improving their reusability and sample-efficiency. We next evaluate our proposed method on two challenging multi-agent domains (§5). Our complete method outperforms both state-of-the-art hierarchical and standard MARL methods as well as strong allocation heuristics. Furthermore, we evaluate the practicality of our subtask action-value function decomposition, demonstrating that learning fails without it while highlighting cases in which the required assumptions do not hold. Lastly, we show the importance of learning action policies in concert with allocation policies.

## 2 Related Work

**Multi-Agent Task Allocation**    *Task allocation* is a long-studied problem in multi-agent systems and robotics [26, 9, 12, 8]. Given a set of tasks and a set of agents, the goal in this setting is to assign agents to tasks in a manner that maximizes total utility. A utility function which maps agent team-task pairs to their utility is assumed to be given. Within the taxonomy of sub-classes to this problem formalized by Gerkey and Matarić [9], our problem setting can be classified as ST-MR-IA (single-task agents, multi-agent tasks, instantaneous allocation). In other words, agents cannot multi-task, tasks require the cooperation of several agents, and they are assigned instantaneously without incorporating any information regarding future task availability. We note that work in multi-agent task allocation often assumes the *execution* of tasks to be trivial; the focus is placed entirely on how to assign agents to a set of tasks in an efficient manner, and some knowledge of agents' effectiveness in each task is assumed. In our reinforcement learning setting, we only assume the ability to execute actions in an environment and observe global and subtask-specific rewards. Our focus is on enabling the application of MARL to complex composite tasks, rather than learning task allocation policies that compete with traditional approaches.

**MDP Formulations for Multi-Agent Task Allocation**    Several works have proposed formulations of task allocation as an MDP, enabling the use of learning and planning-based approaches. Proper and Tadepalli [20] introduce the decomposition of the simultaneous task setting into *task allocation* and

*task execution* levels. However, their allocation selection procedure does not consider how allocations may change in the future. Campbell et al. [6] address this shortcoming by taking future allocations into account when allocating agents to subtasks; however, they assume the pre-existence of low-level action policies. Notably, neither approach learns a tractable policy over the allocation space, and instead relies on exhaustive search or hand-crafted heuristics in order to select allocations.

**Multi-Agent Reinforcement Learning**   Several recent works in MARL have addressed related problems to ours. Most relevantly, Carion et al. [7] introduce a learning-based approach for task allocation that scales to complex tasks. Unlike our proposed setting, their work assumes pre-existing subtask execution policies, as well as domain knowledge regarding how much an agent "contributes" to subtasks, which are used as constraints in a linear/quadratic program to prevent assigning more agents than necessary to subtasks. Our method, on the other hand, learns low-level policies, assumes less prior knowledge, and solves the problem of finding the best allocation by *learning* an allocation controller from experience rather than using an off-the-shelf solver with inputs generated by a learned function and constraints from domain knowledge.

Shu and Tian [25] devise a method whereby a central "manager" agent provides *incentives* to self-interested "worker" agents to perform specific subtasks such that the manager's reward is maximized and incentives provided to workers are minimized. In this case, agents work on subtasks independently and are not cooperating towards a shared goal. Yang et al. [37] also consider a setting where multiple simultaneous tasks (described as *goals*) exist in a shared environment; however, they assume that goals are pre-assigned to agents and therefore do not tackle the task allocation problem. Liu et al. [15] introduce COPA, a hierarchical MARL method which learns a centralized controller in order to coordinate decentralized agents. Unlike our method, which utilizes task allocation for the purpose of decomposing complex settings to simplify learning, COPA's motivation is primarily to alleviate the problem of partial observability during decentralized execution. We evaluate our method against COPA in order to assess the effectiveness of our hierarchical decomposition in comparison to a more generic hierarchical MARL method with similar assumptions (i.e. no pre-trained policies, cooperative tasks, and no pre-existing task allocation).

## 3   Preliminaries

**Task Framework**   Our setting builds on the general framework of Decentralized POMDPs [19] with entities [23, 10] by incorporating structure for multiple coexisting subtasks. Note that we differentiate between subtasks and tasks. We refer to independent jobs within an environment as "subtasks" and the global collection of them (specified by the Dec-POMDP) as a task. This framework describes fully cooperative problems which can be specified by the tuple: $(\mathbf{S}, \mathbf{U}, \mathbf{O}, P, r, (r_i, \mathcal{E}_i \,|\, i \in \mathcal{I}), \mathcal{A}, \mu, \mathcal{I})$. $\mathcal{A}$ is the set of agent entities. Each entity $e$ has a state $s^e$. We denote, for example, the state space of all agents as $\mathbf{S}_{\mathcal{A}}$ where $\{s^a | a \in \mathcal{A}\} \in \mathbf{S}_{\mathcal{A}}$. The state space of other sets of entities is denoted similarly. Each agent $a$ can execute actions $u^a$, and the joint action of all agents is denoted $\mathbf{u} = \{u^a | a \in \mathcal{A}\} \in \mathbf{U}$. Within the environment exist a set of subtasks $\mathcal{I}$ where each is defined by a set of subtask-specific entities $\mathcal{E}_i$ and the subtask reward function $r_i : \mathbf{S}_{\mathcal{E}_i} \times \mathbf{S}_{\mathcal{A}} \times \mathbf{U} \to \mathbb{R}$. The relevant state (i.e. set of entity states) for subtask $i$ is then denoted as $s_i = \{s^e | e \in \mathcal{E}_i \cup \mathcal{A}\}$. We define $\mathcal{E}$ as the set of all entities (including agents) in the environment: $\mathcal{E} := \left(\bigcup_{i \in \mathcal{I}} \mathcal{E}_i\right) \cup \mathcal{A}$. The global state is the set $\mathbf{s} := \{s^e \,|\, e \in \mathcal{E}\} \in \mathbf{S}$. Not all entities may be visible to each agent, so we define a binary observability mask: $\mu(s^a, s^e) \in \{1, 0\}$. Thus, an agent's observation is defined as $o^a = \{s^e | \mu(s^a, s^e) = 1, e \in \mathcal{E}\} \in \mathbf{O}$. $P$ is the state transition distribution which defines the probability $P(\mathbf{s}' | \mathbf{s}, \mathbf{u})$. Finally, we define the global reward function $r : \mathbf{S} \times \mathbf{U} \to \mathbb{R}$. In the simplest case, the global reward function can be a sum of the individual subtask rewards; however, it can optionally encode global objectives (e.g. a scalar reward upon the completion of *all* subtasks).

**Cooperative MARL via Value Function Factorization**   Cooperative MARL methods are concerned with learning a set of policies which maximize the expected discounted sum of global rewards. Value function based methods do this by learning an approximation of the optimal $Q$-function:

$$Q^{\text{tot}}(\mathbf{s}, \mathbf{u}) := \mathbb{E}\left[ \sum_{t=0}^{\infty} \gamma^t \, r(\mathbf{s}_t, \mathbf{u}_t) \,\middle|\, \begin{smallmatrix} \mathbf{s}_0 = \mathbf{s}, \ \mathbf{u}_0 = \mathbf{u}, \ \mathbf{s}_{t+1} \sim P(\cdot | \mathbf{s}_t, \mathbf{u}_t) \\ \mathbf{u}_{t+1} = \arg\max Q^{\text{tot}}(\mathbf{s}_{t+1}, \cdot) \end{smallmatrix} \right]$$

$$= r(\mathbf{s}, \mathbf{u}) + \gamma \, \mathbb{E}\left[ \max Q^{\text{tot}}(\mathbf{s}', \cdot) \,|\, s' \sim P(\cdot | \mathbf{s}, \mathbf{u}) \right]$$

An optimal policy can be derived from this $Q$-function by taking the maximum valued action $\boldsymbol{u}$ from the current state; however, in the multi-agent setting it is often desirable for each agent to execute their policy independently (i.e. in a decentralized fashion). In order to derive decentralized

policies, the $Q$-function is represented in a manner that realizes the Individual-Global-Max (IGM) principle [27] which states that each agent can greedily maximize their local utility $Q^a$ and maximize $Q^{\text{tot}}$ as a result. This has been accomplished through representing $Q$ as a summation of utilities [28], a monotonically increasing combination [21] of them, as well as several works which attempt to extend the representational capacity of the global $Q$-function to *all* functions satisfying the IGM principle [27, 36]. The value function approximation is learned by training a neural network to minimize the following loss derived from the Bellman equation:

$$\mathcal{L}_Q(\theta) := \mathbb{E}\left[\left(y_t^{\text{tot}} - Q_\theta^{\text{tot}}(\boldsymbol{s}_t, \mathbf{u}_t)\right)^2 \Big|_{(\boldsymbol{s}_t, \mathbf{u}_t, r_t, \boldsymbol{s}_{t+1}) \sim \mathcal{D}}\right]$$
$$y_t^{\text{tot}} := r_t + \gamma Q_{\bar{\theta}}^{\text{tot}}\left(\boldsymbol{s}_{t+1}, \arg\max Q_\theta^{\text{tot}}(\boldsymbol{s}_{t+1}, \cdot)\right) \tag{1}$$

where $\bar{\theta}$ are the parameters of a target network that is copied from $\theta$ periodically to improve stability [17] and $\mathcal{D}$ is a replay buffer [14] that stores transitions collected by an exploratory policy (typically $\epsilon$-greedy). In cases of partial observability, the history of local actions and observations $\tau_t^a = (o_1^a, u_1^a, \ldots, o_t^a)$ is used as a proxy for the state $\boldsymbol{s}_t$ as input to the agents' utility functions $Q^a$.

**Hierarchical MARL**   Value-based single-agent Hierarchial RL methods [e.g. 13] typically learn a temporally abstracted high-level $Q$-function which selects goals upon which a low-level controller is conditioned. In adapting these methods for the multi-agent setting, one can choose to select goals in a fully decentralized fashion [29] or through a centralized controller [15, 38, 1]. Our work will consider the latter setting, where a single controller learns a value function over a shared goal space $\boldsymbol{b} \in \boldsymbol{B}$:

$$Q(\mathbf{s}, \mathbf{b}) = \sum_t^{N_t} r_t + \gamma \mathbb{E}\left[\max Q(\mathbf{s}', \cdot) | \mathbf{s}' \sim P, \pi_{\mathbf{b}}\right] \tag{2}$$

This controller operates over a dilated time scale; one step from its perspective amounts to $N_t$ steps in the environment executed by low-level controllers conditioned on the goal, $\pi_{\boldsymbol{b}}$. The low-level controllers are trained to maximize goal-specific rewards which can be manually defined or learned.

## 4   Allocator-Actor Multi-Agent Architecture

In the last section we introduced a generic framework for value-based hierarchical MARL. Now we will present the details specific to our hierarchical MARL framework for composite tasks using subtask allocation, **ALMA**. In this case we define the action space of the high-level controller (whose value function is defined in Eqn. 2) as the set of all possible allocations of agents to subtasks. A *subtask allocation* refers to a specific set of agent-subtask assignments, where each agent can only be assigned to one subtask at a time, and it is denoted by $\mathbf{b} = \{b^a \mid a \in \mathcal{A}\} \in \mathbf{B}$, where $b^a \in \mathcal{I}$ represents the subtask that agent $a$ is assigned to. We then slightly abuse notation and denote the set of agents assigned to subtask $i$ as $\boldsymbol{b}_i \subseteq \mathcal{A}$ and the set of all agents *not* assigned to task $i$ as $\boldsymbol{b}_{\backslash i}$ where $\boldsymbol{b}_i \cup \boldsymbol{b}_{\backslash i} = \mathcal{A}$. We then specify the joint action for agents assigned to subtask $i$ as $\mathbf{u}_{\mathbf{b}_i} = \{u^a | a \in \boldsymbol{b}_i\}$. Recall that the subtask-specific state $\boldsymbol{s}_i$ includes the state of *all* agents. We denote the subtask-specific state including *only* the assigned agents as $\boldsymbol{s}_{\boldsymbol{b}_i} = \{s^e | e \in \mathcal{E}_i \cup \boldsymbol{b}_i\}$.

### 4.1   Subtask Allocation Controllers

One of the main challenges of learning a $Q$-function for the high-level allocation controller is the massive action space. Our formulation of subtask allocation can be seen as a set partitioning problem, which is known to be NP-Hard [24, 9]. $Q$-Learning with the allocation action space requires finding the allocation with the highest $Q$-value at each step (Eqn. 2), as does deriving an action policy from this $Q$-function. At each step, we have a choice of $|\mathcal{I}|^{|\mathcal{A}|}$ possible unique allocations of agents to subtasks, and it is prohibitively expensive to evaluate each one and select the best. Van de Wiele et al. [31] introduce "Amortized $Q$-Learning" which addresses the problem of massive action spaces in Deep $Q$-Learning by defining a "proposal distribution" which they train to maximize the density of actions with high values given a state. We adapt this idea for our allocation controller, where $f(\boldsymbol{b}|\boldsymbol{s}; \phi)$ is our proposal distribution over allocations. We can then sample from this distribution and select the allocation with the highest value, effectively approximating the maximization procedure required for $Q$-Learning. Our proposal distribution is learned with the following loss:

$$\mathcal{L}(\phi; \boldsymbol{s}) = -\log f(\boldsymbol{b}^*(\boldsymbol{s})|\boldsymbol{s}; \phi) - \lambda^{\text{AQL}} H(f(\cdot|\boldsymbol{s}; \phi)), \tag{3}$$

where $\boldsymbol{b}^*(\boldsymbol{s})$ is the highest-valued allocation from a set of $N_p$ samples from the proposal distribution. Formally, $\boldsymbol{b}^*(\boldsymbol{s}) := \arg\max_{\boldsymbol{b} \in \boldsymbol{B}^{\text{samp}}(s)} Q(\boldsymbol{s}, \boldsymbol{b})$ where $\boldsymbol{B}^{\text{samp}}(s) := \{\boldsymbol{b}^1, \ldots, \boldsymbol{b}^{N_p} \sim f(\cdot|\boldsymbol{s}; \phi)\}$. We

then learn an approximation of Eq. 2 with the following loss, adapted from Eqn. 1:

$$\mathcal{L}_Q(\Theta) := \mathbb{E}\left[\left(y_t - Q_\Theta(\boldsymbol{s}_t, \mathbf{b}_t)\right)^2\right], \quad y_t := \sum_n^{N_t} r_{t+n} + \gamma Q_{\bar{\Theta}}\left(\boldsymbol{s}_{t+N_t}, \boldsymbol{b}^*(\boldsymbol{s}_{t+N_t})\right) \qquad (4)$$

where transitions $(\boldsymbol{s}_t, \mathbf{b}_t, \sum_n^{N_t} r_{t+n}, \boldsymbol{s}_{t+N_t})$ are sampled from a replay buffer. Learning a proposal distribution and value function over a combinatorial space closely parallels recent work in learned combinatorial optimization [3, 11, 4].

We construct our proposal distribution modularly so that each module can leverage the fact that subtasks are drawn from a shared distribution. These modules consist of an agent embedding function $f^h : \boldsymbol{S}_a \to \mathbb{R}^d$, a subtask embedding function $f^g : \mathbf{S}_{\mathcal{E}_i} \to \mathbb{R}^d$, and a subtask embedding update function $f^u : \mathbb{R}^{2d} \to \mathbb{R}^d$ which is used to update the embedding of the subtask that an agent was assigned to. We first construct $f$ in a factorized auto-regressive form: $f(\mathbf{b}|\mathbf{s}) = \prod_{a \in \mathcal{A}} f(b^a|\mathbf{s}, \mathbf{b}^{<a})$, where $\mathbf{b}^{<a}$ are the allocations made prior to assigning agent $a$. Thus, we are iteratively assigning agents to subtasks while accounting for allocations already determined.

Since the number of subtasks present at any time is variable (e.g. some subtasks may be finished before others), we construct each auto-regressive factor as a composition of outputs generated by the modules described above, in a fashion similar to pointer-networks [34]. Pointer networks are commonly used in RL applications to handle variable-sized action spaces where the individual actions correspond to entities/groups [35]. We first generate a set of subtask embeddings $\boldsymbol{g}_i = f^g(\boldsymbol{s}_{\mathcal{E}_i})$ encoding the state of the non-agent entities belonging to each subtask, as well as embeddings describing each agent's state $\boldsymbol{h}_a = f^h(s^a)$. Logits are computed as the dot-product between agent embeddings and subtask embeddings: $f(b^a|\mathbf{s}, \mathbf{b}^{<a}) = \text{softmax}_{i \in \mathcal{I}}\left(\boldsymbol{g}_i^\top \boldsymbol{h}_a\right)$. We then sample $b^a \sim f$, and update the selected subtask's embedding as $\boldsymbol{g}'_{b^a} = \boldsymbol{g}_{b^a} + f^u(\boldsymbol{g}_{b^a}, \boldsymbol{h}_a)$ such that other agents' allocations can take into account existing ones.

## 4.2 Subtask Execution Controllers

Given a subtask allocation $\boldsymbol{b}$, we must learn low-level action-value functions as described in Section 3. Specifically, for each team $\boldsymbol{b}_i$ we learn a value function based on the subtask rewards $r_i$. Recall that the subtask reward function depends on *all* agents' actions and states since any agent can potentially contribute to any subtask. From the perspective of the team assigned to subtask $i$, we treat the other subtasks' agents as part of the environment by taking the expectation over actions sampled from their optimal policies:

$$Q_i^{\text{tot}}(\mathbf{s}, \mathbf{u}_{\mathbf{b}_i}; \boldsymbol{b}) = \mathbb{E}\left[r_i(\boldsymbol{s}_i, \boldsymbol{u}) + \gamma \max Q_i^{\text{tot}}(\boldsymbol{s}', \cdot; \boldsymbol{b}) \,\middle|\, \begin{matrix} \boldsymbol{u}_{\mathbf{b}\backslash i} \sim \pi^*_{\mathbf{b}\backslash i} \\ \mathbf{s}' \sim P(\cdot|\boldsymbol{s}, \boldsymbol{u}) \end{matrix} \right] \qquad (5)$$

While this function depends on the full global state since all entities can feasibly influence all other entities' state transitions and any agent can potentially contribute to the rewards of any subtask, in composite tasks the set of truly relevant information may be much smaller for optimal policies. Reconsider the example from Sec. 1 where firefighters are agents and buildings are subtasks. While it is possible for any firefighter to put out any fire, given optimal policies we expect firefighters to only put out the fire they are assigned to. Moreover, in the optimal setting, firefighters will not be directly interacting with firefighters at other buildings. As such, for composite tasks, we can decompose the environment into a set of independent sub-environments and rewrite Eqn. 5 as follows:

$$Q_i^{\text{tot}}(\mathbf{s}_{\mathbf{b}_i}, \mathbf{u}_{\mathbf{b}_i}; \boldsymbol{b}) = r_i^{\boldsymbol{b}}(\boldsymbol{s}_{\mathbf{b}_i}, \boldsymbol{u}_{\mathbf{b}_i}) + \gamma \mathbb{E}\left[\max Q_i^{\text{tot}}(\mathbf{s}'_{\mathbf{b}_i}, \cdot; \boldsymbol{b})\big|\mathbf{s}'_{\mathbf{b}_i} \sim P_i^{\boldsymbol{b}}(\cdot|\mathbf{s}_{\mathbf{b}_i}, \boldsymbol{u}_{\mathbf{b}_i})\right] \qquad (6)$$

where $P_i^{\boldsymbol{b}}$ is the transition distribution of sub-environment $i$ and the transitions of the full environment can be written in the following factored form: $P(\boldsymbol{s}'|\boldsymbol{s}, \boldsymbol{u}) = \prod_{i \in \mathcal{I}} P_i^{\boldsymbol{b}}(\boldsymbol{s}'_{\boldsymbol{b}_i}|\boldsymbol{s}_{\boldsymbol{b}_i}, \boldsymbol{u}_{\boldsymbol{b}_i}) \forall \boldsymbol{s}, \boldsymbol{s}' \in \boldsymbol{S}_{\boldsymbol{b}}^*$. Given subtask allocation $\boldsymbol{b}$, $\boldsymbol{S}_{\boldsymbol{b}}^*$ is the set of global states visited by the optimal subtask policies, $\pi^*_{\boldsymbol{b}_i}$. Finally, $r_i^{\boldsymbol{b}}$ is the sub-environment reward function that depends only on local information defined as: $\exists r_i^{\boldsymbol{b}} : \boldsymbol{S}_{\mathcal{E}_i} \times \boldsymbol{S}_{\boldsymbol{b}_i} \times \boldsymbol{U}_{\boldsymbol{b}_i} \to \mathbb{R}$ s.t. $r_i^{\boldsymbol{b}}(\boldsymbol{s}_{\boldsymbol{b}_i}, \boldsymbol{u}_{\boldsymbol{b}_i}) = r_i(\boldsymbol{s}_i, \boldsymbol{u}) \forall \boldsymbol{s} \in \boldsymbol{S}_{\boldsymbol{b}}^*, \boldsymbol{u} \in \boldsymbol{U}$.

Note that the $Q$-function now only depends on a "local" state, and, as such, can be approximated more easily due to the smaller function input space. Moreover, in the case where subtasks are drawn from the same distribution, $Q$-functions defined over the local state will more easily generalize to other instantiations of the same subtask since they no longer depend on the context outside of that subtask's state. We validate the usefulness of this decomposition empirically in Section 5.

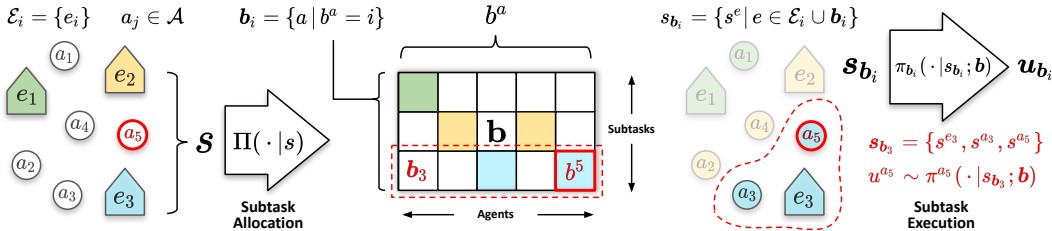

Figure 2: **ALMA** computing agent actions (specifically agent $a_5$) given the current state. Subtask allocations $\boldsymbol{b}$ are updated by a centralized controller $\Pi$ every $N_t$ steps, and then agent policies $\boldsymbol{\pi}$ select low-level actions $\boldsymbol{u}$ in a decentralized fashion given their local state.

Learning a function that approximates Eqn. 6 requires predicting $Q$-values for teams of varying sizes, as the quantity of agents assigned to a subtask is not fixed. In fact, each unique combination of agents assigned to subtask $i$ can be seen as a unique Dec-POMDP. We rely on recent work [10] which learns factorized multi-task $Q$-functions for teams of varying sizes by sharing parameters via attention mechanisms. This work falls under the category of factorized value function methods for cooperative MARL described in §3, and as such, we represent each subtask-specific $Q$-function as a monotonic mixture of agent utility functions computed in a decentralized fashion, such that agents can act independently without communication.

### 4.3   Training and Execution Details

We provide an overview of the complete subtask allocation and execution procedure in Figure 2. Subtask allocations $\boldsymbol{b}$ are selected in a centralized fashion every $N_t$ steps which then determines the set of low level policies to execute. We define the high-level policy $\Pi(\boldsymbol{b}|\boldsymbol{s}) := \mathbb{1}(\boldsymbol{b} = \boldsymbol{b}^*(\boldsymbol{s}))$ where $\boldsymbol{b}^*(\boldsymbol{s})$ is the highest valued allocation sampled from the proposal distribution, as defined in Section 4.1. The per-agent low-level policy for agent $a$ assigned to subtask $i$ is defined as $\pi^a(u^a|\boldsymbol{s}_{\boldsymbol{b}_i};\boldsymbol{b}) := \mathbb{1}(u^a = \arg\max_{u^{a\prime}} Q_i^a(\boldsymbol{s}_{\boldsymbol{b}_i}, u^{a\prime};\boldsymbol{b}))$ where $Q_i^a$ is the agent's utility function which is monotonically mixed with all $a \in \boldsymbol{b}_i$ to form $Q_i^{\text{tot}}$.

When collecting data during training, we select random low-level actions with probability $\epsilon$ in order to promote exploration. We use two types of exploration in the allocation controller. With probability $\epsilon^p$ we select a full allocation sampled from the proposal distribution, rather than taking the highest valued one. Then, with probability $\epsilon^r$, we randomly select subtask allocations *independently* on a per-agent basis. All exploration probabilities are annealed over the course of training, and the annealing schedules (along with all hyperparameters) are provided in Appendix D.

To handle state spaces of variable size (i.e. variable quantity of entities), we use attention models [33] as in [10] for all components. Network architectures are provided in Appendix A. In order to achieve the partial views over the global state required by our redefined low-level $Q$-function in Eqn. 6, we use masking in attention models to prevent agents from seeing certain entities.

## 5   Experiments

We evaluate **ALMA** on two challenging environments, described below (more in Appendix B).

### 5.1   Environments

**SAVETHECITY**    Inspired by the classical Dec-POMDP task "Factored Firefighting" [18], and based on the scenario described in Sec. 1, we introduce SAVETHECITY, an environment where agents must coordinate to extinguish fires and rebuild a burning city (see Fig. 3). In this version, agents are capable of contributing to any subtask, must physically navigate between buildings through low-level actions, and there are different agent types each with distinct abilities that are crucial for success. The subtask allocation function must discover these abilities from experience and use this knowledge to distribute agents effectively. The task also shares similarities with the Search-and-Rescue task from [15], but we do not consider partial observability and we introduce agents with diverse capabilities.

**STARCRAFT**    Our next environment is the StarCraft multi-agent challenge (SMAC) [22] which involves a set of agents engaging in a battle with enemy units in the StarCraft II video game. We specifically consider the multi-task setting presented by Iqbal et al. [10] where each task presents a unique combination and quantity of agent and enemy types. As such, agents cannot learn fixed

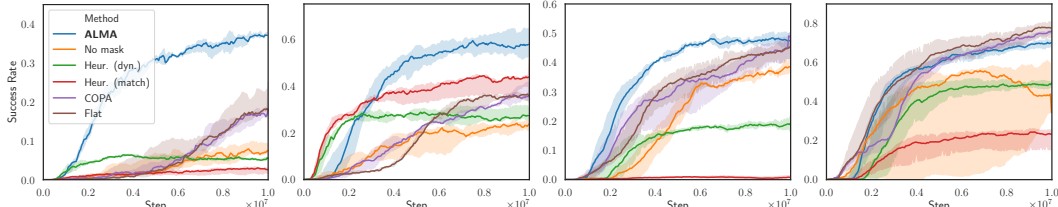

Figure 4: STARCRAFT II training curves, left to right: S&Z (a) disadvantage and (b) symmetric; MMM (c) disadvantage and (d) symmetric. Shaded region is a standard deviation across 5 seeds.

strategies, and must instead adapt strategies to their set of teammates and enemies. While the original setting consists of a single enemy army attacking the agents, inspired by the full game of StarCraft we increase the complexity by introducing multiple enemy armies (see Fig. 5) which periodically attack and retreat independently and the additional objective of defending a base. By doing do, we are essentially composing several tasks of similar complexity to those considered by the current literature and requiring agents to learn not only how to solve those tasks but also how to distribute agents across tasks in a manner that is globally optimal. Within this setting, we consider four different variations of army compositions based on those presented in [10]. In one set of settings we give agents and enemies a symmetric set of units, while in the other, agents have one fewer unit than the enemies. Within each set we consider two different pools of unit types: "Stalkers and Zealots" (S&Z) and "Marines, Marauders, and Medivacs" (MMM). The former includes a mixture of melee and ranged units, while the latter includes units that are capable of healing.

## 5.2 Results

In our experimental validation we aim to answer the following questions: 1) *Learning Efficacy:* Is **ALMA** effective in improving learning efficiency and asymptotic performance in comparison to state-of-the-art hierarchical and non-hierarchical MARL methods? 2) *Allocation Strategic Complexity*: Are the learned allocation policies non-trivial or are any benefits of **ALMA** purely gained from the subtask decomposition? 3) *Decomposition Validity*: What, if anything, do we gain from the sub-environment and how does **ALMA** fare when the required assumptions are broken? 4) *Joint Training*: Do we receive any benefits from training allocation and execution controllers jointly?

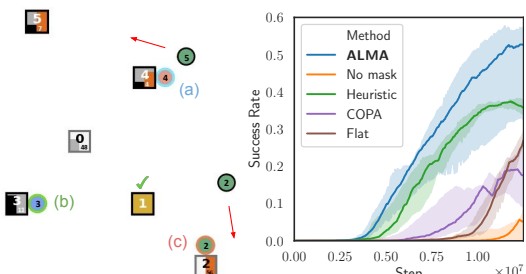

Figure 3: SAVETHECITY example task and training curves. Shaded region is a standard deviation across 8 seeds.

**Learning Efficacy**    First, we aim to evaluate the effectiveness of our hierarchical abstraction in comparison to state-of-the-art cooperative MARL methods in variable entity multi-task settings. We compare to two non-hierarchical multi-agent baselines: QMIX [21] augmented with self-attention [33], to extract information from variable quantities of entities (A-QMIX), and REFIL [10], a state-of-the-art approach built on top of A-QMIX for generalizing across tasks with different compositions of agent and entity types. Both methods are referred to as "Flat" in our figures and differ depending on the environment. REFIL is used for STARCRAFT tasks, but we find that it does not significantly improve on A-QMIX in SAVETHECITY, so we use A-QMIX in that setting. Evaluating these methods on our tasks allows us to isolate the effectiveness of our hierarchical decomposition, as we use the same algorithms to train low-level controllers in ALMA.

Next, we compare to a state-of-the-art approach in hierarchical MARL for varying entity settings: COPA [15]. COPA makes the assumption, similar to **ALMA**, of a centralized agent that is able to communicate with decentralized agents periodically. We ensure that the communication frequency of COPA matches that of our method and that it is provided subtask labels for entities, so it receives a similar amount of information, even though it does not explicitly leverage the subtask decomposition.

In both SAVETHECITY and STARCRAFT, we find that **ALMA** is able to outperform all baselines in most settings. Both A-QMIX (Flat) and COPA are unable to converge to a reasonable policy within the allotted timesteps in SAVETHECITY (Fig. 3), and REFIL (Flat) and COPA are only competitive with **ALMA** in terms of both sample efficiency and asymptotic performance in one setting: MMM

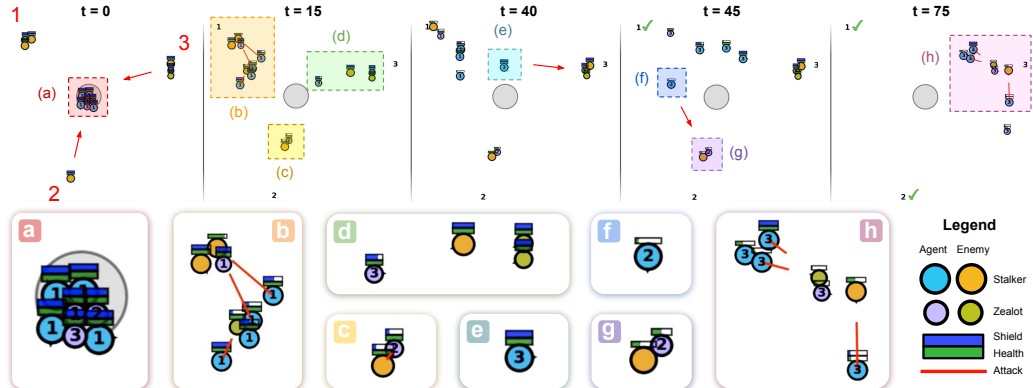

Figure 5: Walk-through of learned **ALMA** policy on a sample STARCRAFT task. Agents start ($t = 0$) at the center, and two enemy armies immediately begin to attack (a). **ALMA** learns there is an advantage in numbers (e.g. "focus firing" on single enemies) and allocates most agents to Subtask 1 (b). One Zealot is additionally assigned to each attacking army to prevent the base from being stormed (c-d). This strategy was determined from the very first time-step (a). Around $t = 40$, a Stalker is reallocated to Subtask 3 (e), as the Zealot previously assigned was defeated, leaving a vulnerability, and Subtask 1 has almost been completed by the other agents. Once Subtask 1 is complete ($t = 45$), most agents are allocated to Subtask 3, the largest army, but one Stalker (f) is allocated to assist the Zealot at Subtask 2 (g) since it is low on health. Once Subtask 2 is complete, the remaining agents are allocated to Subtask 3 to complete the task.

Symmetric (Fig. 4d). These results highlight the importance of **ALMA**'s hierarchical abstraction as well as the sub-environment decomposition used in order to accelerate learning of low-level policies.

Interestingly **ALMA** appears to supply the greatest performance gains on the environments that are most difficult, as judged by the absolute success rates. **ALMA** stands out in S&Z disadvantage (Fig. 4), where the next best methods achieve 20% success, while it only roughly matches the performance of the top performing methods in MMM symmetric where the highest success rates are around 75%. We hypothesize that when agents are faced with such a disadvantage, coordinated strategy becomes more crucial.

**Allocation Strategic Complexity**  Next, we hope to learn whether our allocation controller is learning complex non-trivial strategies. As such, we implement allocation heuristics derived from domain knowledge in each setting and only learn the low-level controllers. These methods serve as a strong baseline for learned allocation, as they simplify the learning problem for low-level controllers by leveraging the sub-environment decomposition (i.e. masking subtask-irrelevant entities) and having a fixed high-level allocation strategy. For SAVETHECITY, the heuristic allocates each agent to the nearest building at which they are most useful according to their capabilities. While relatively simple, devising a more sophisticated allocation strategy is nontrivial as there are many factors to weigh. For STARCRAFT, we devise two heuristics: the first (*matching*) allocates agents to enemy armies by matching unit types such that each individual battle is fair. The other (*dynamic*) only considers enemy armies that are currently attacking and also attempts to match the unit composition.

In SAVETHECITY (Fig. 3) we find the heuristic is able to learn quickly at the beginning as the action-level policies improve, but it converges lower than **ALMA**. In STARCRAFT (Fig. 4) we find that at least one of the heuristics can be competitive in some cases (e.g. S&Z Symmetric), although which heuristic performs best is not consistent across settings, and **ALMA** always outperforms it. With these results we can conclude decomposing the task into subtasks to be solved by simpler task execution controllers is not sufficient for superior performance in our environments and sophisticated high-level strategy must be learned.

In Fig. 5 we qualitatively demonstrate some of the strategies that **ALMA** is able to learn. We find that **ALMA** is able to allocate agents to subtasks in a manner that considers long term consequences and balances priorities gracefully (e.g. defending the base vs. defeating enemies).

**Decomposition Validity**  In order to validate the sub-environment decomposition introduced in Eqn. 6 we ablate our approach to exclude the task-specific masking performed on each agent's observation. This method is referred to as "No mask" in Figures 3 and 4, where we find that our method experiences a significant deterioration in performance when not masking irrelevant information.

To test the boundaries of the required assumptions (independent transitions), we evaluate on settings in STARCRAFT which violate them. While we train the agents only in cases where enemy armies are maximally spread out from each other around the agents' base, in our evaluation setting we allow for armies to spawn arbitrarily close to one another. In this case, for example, agents assigned to one army may be attacked by other armies or bump into agents attacking other armies, violating subtask transition independence. We plot the performance of **ALMA** alongside REFIL (Flat) in Fig. 6a, where the distance between armies varies along the x-axis. The further right, the more similar the tasks are to what is seen during training.

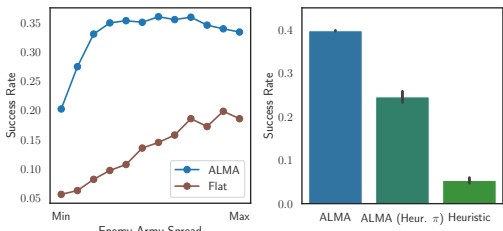

Figure 6: Further analysis in STARCRAFT S&Z disadvantage setting. Left (a): Varying spread of enemy armies. Right (b): Zero-shot evaluation with alternate action policies.

While REFIL's performance deteriorates smoothly as the composition of subtasks become more dissimilar to those seen during training, **ALMA** maintains steady performance and even sees a slight uptick in performance towards the middle. While surprising, this difference in generalization capability can be attributed to the modularity of our approach. While REFIL treats each unique composition of subtasks as novel, the modules comprising our approach (specifically the subtask-execution controllers and allocation proposal network) only see information relevant to their subtask, which are individually drawn from the same distribution as seen during training. **ALMA**'s increase in performance in the middle can be attributed to the fact that armies being closer together makes it easier for agents to switch between tasks. Ultimately, as armies become closer together, our assumptions begin to fail, subtasks blur together, and we finally see **ALMA**'s performance drop off.

**Joint Training**   Although **ALMA** is able to learn all components (subtask allocation and execution/action controllers) jointly, we evaluate its ability to utilize agent policies trained by a different method in a zero-shot manner. In particular, we see what happens when we replace **ALMA**'s action-level policies with those trained with a heuristic. We note that the procedure for training the low-level controllers is identical between these methods and the only difference is the allocation strategy used throughout training. We see in Figure 6b that without further training, **ALMA**'s superior allocation strategy is able to *quadruple* the effectiveness of the policies learned with the heuristic. Ultimately though, the best performance is only attainable through training allocation and execution controllers jointly, as the execution controllers can learn to succeed in the settings the allocation controllers put them in, and vice versa, creating a virtuous feedback loop.

## 6   Conclusion and Future Work

In this work we have introduced **ALMA**, a general learning method for tackling a variety of multi-agent composite tasks in which each subtasks' rewards and transitions can be assumed as independent. By simultaneously learning a high-level allocation policy and action-level agent policies, **ALMA** is able to succeed in settings which are difficult for flat methods, and in which well-performing heuristics are difficult to come by. One promising direction for future work is to equip **ALMA**'s allocation policy with more sophisticated exploration methods. Although the stochasticity resultant from the $\epsilon$-greedy procedure and the proposal distribution sampling yields surprisingly good results, a more intelligent exploration procedure could even better alleviate the challenges inherent to learning over such a large combinatorial action space. We expect such an improvement would, for example, prevent some of the variance we observe in SAVETHECITY (Figure 3). Another direction for future work is to discover subtasks and/or observation masks from the environment automatically. We believe this work has immediate practical implications in real-world settings that can be described as composite tasks, such as warehouse robotics.

## 7   Acknowledgements

We thank Sébastien Arnold and the anonymous reviewers for their feedback. This work is partially supported by NSF Awards IIS-1513966/ 1632803/1833137, CCF-1139148, DARPA Award#: FA8750-18-2-0117, FA8750-19-1-0504, DARPAD3M - Award UCB-00009528, Google Research Awards, gifts from Facebook and Netflix, and ARO# W911NF-12- 1-0241 and W911NF-15-1-0484.

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
