# Appendix

## A   Network Architectures

Our allocation proposal network and $Q$ network are illustrated in Figures 7 and 8. Low-level action utility functions and mixing networks are similar to those described in Iqbal et al. [10] with the only

difference being a replacement of the RNN layers with standard fully connected layers. We omit RNNs, as we do not consider settings with significant partial observability. Furthermore, the run time is significantly faster since all time steps can be batched during training. We find the omission of RNNs does not affect performance in the domains initially used by Iqbal et al. [10].

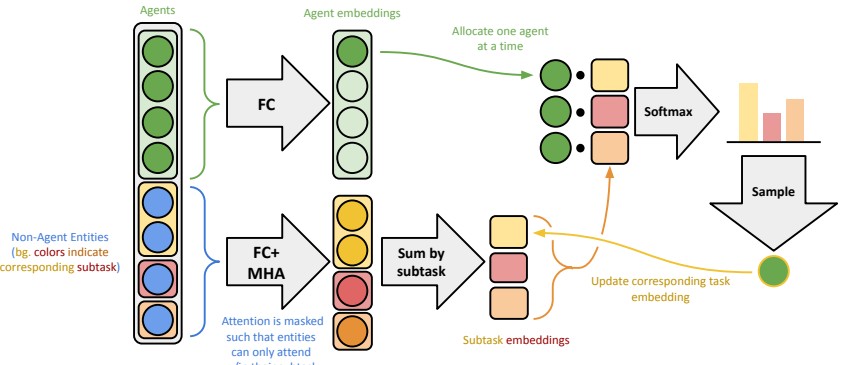

Figure 7: Allocation proposal network structure as described in §4.1

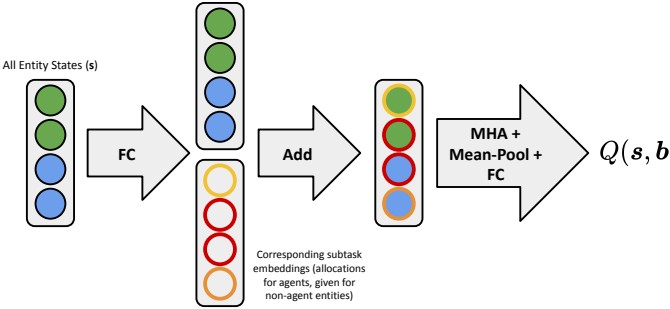

Figure 8: Allocation $Q$-function network structure.

## B   Environment details

SAVETHECITY    This environment is inspired by the classical Dec-POMDP task "Factored Fire-fighting" [18] where several firefighters must cooperate to put out independent fires; however, we introduce several additional degrees of complexity. First, agents are capable of contributing to *any* subtask, such that the task is amenable to subtask allocation, and we can not use a fixed value function factorization. Second, agents are embodied and must physically move themselves to buildings through low-level actions, rather than being fixed and only having a high-level action space selecting buildings to fight fires at. Finally, we introduce several types of agents with differing capabilities (all of which are crucial for success), such that the subtask allocation function must learn which subtasks require which capabilities and how to balance these. The task also shares similarities with the Search-and-Rescue task from [15]; however, we do not consider partial observability and we introduce agents with diverse capabilities.

In each episode, there are $N = [2, 5]$ agents (circles) and $N + 1$ buildings (squares) (see 3). Each building regularly catches fire (red bar) which reduces the building's "health" (black bar). The agents must learn to put out the fires and then fully repair the damage, at which point the building will no longer burn. The episode ends when all buildings are fully repaired or burned down, and an episode is considered successful if no buildings burn down. The *firefighter* (red) and *builder* (blue) agents are most effective at extinguishing fires (a) and repairing damaged buildings (b), respectively, while *generalist* (green) agents—though unable to make progress on their own—can move twice as fast, prevent further damage to a blazing building (c), and increase the effectiveness of other agents at their weak ability if at the same building.

The full map is a 16x16 grid and buildings are randomly spawned across the map. Agents always start episodes in a cluster in the center. Buildings begin episodes on fire at a 40% rate. Agents are rewarded for increasing a building's health, completing a building, putting out a fire, and completing all buildings (global reward only). Agents are penalized for a building burning down or its health decreasing due to fire.

STARCRAFT    The StarCraft multi-agent challenge (SMAC) [22] involves a set of agents engaging in a battle with enemy units in the StarCraft II video game. We consider the multi-task setting presented by Iqbal et al. [10] where each task presents a unique combination and quantity of agent and enemy types. We increase the complexity by introducing *multiple* enemy armies and the additional objective of defending a centralized base. These enemy armies attack the base from multiple angles, and the agents must defeat all enemy units while preventing any single enemy unit from entering their base in order to succeed. As we do for SAVETHECITY, we train simultaneously on tasks with variable types and quantities of agents.

We consider four settings based on those presented in [10] consisting of unique combinations of unit types which require varying strategies in order to succeed. In one set of settings we give agents and enemies a symmetric (i.e. matching types) set of units, while in the other agents have one fewer unit than the enemies. Within each set we consider two different pools of unit types: "Stalkers and Zealots" (S&Z) and "Marines, Marauders, and Medivacs" (MMM). The former includes a mixture of melee and ranged units, while the latter includes units that are capable of healing. In each setting, there can be between 3 and 8 agents and 2 or 3 enemy armies. Each enemy army consists of up to 4 units. Stalkers are units that are capable of shooting enemies from afar and are useful for causing damage without taking as much damage, as they can run immediately after shooting. Zealots are melee units (i.e. they must walk up to their enemies to damage them), and they are especially strong against Stalkers. Marines are ranged units that are relatively weak with respect to damage output and health. Marauders are also ranged units and have more health and damage output. Medivacs are ships that float above the battlefield and are able to heal their friendly units. Notably, multiple Medivacs cannot heal the same unit simultaneously, so agents can overpower a Medivac's healing by targeting the same unit.

Agents are rewarded for damaging enemy agents' health, defeating enemy agents, and defeating enemy armies. The global reward also includes an additional reward for defeating *all* armies.

## C    Experimental Details

Our experiments were performed on a desktop machine with a 6-core Intel Core i7-6800K CPU and 3 NVIDIA Titan Xp GPUs, and a server with 2 16-core Intel Xeon Gold 6154 CPUs and 10 NVIDIA Titan Xp GPUs. Each experiment is run with 8 parallel environments for data collection and a single GPU.

## D    Hyperparameters and Implementation Details

See Table 1 for an overview of implementation details. We use Pop-Art [32] to normalize returns across varying scales such that we can use similar hyperparameters across environments. Hyperparameters for REFIL [10] and COPA [15] are taken directly from those works. New hyperparameters (e.g. allocation length, epsilon schedules, etc.) were chosen by separately tuning each one. In Figure 9 we see that ALMA is robust to different settings of two allocator-related hyperparamters—even the extreme settings produce results significantly better than the best-performing baseline.

Table 1: Hyperparameter settings across all runs and algorithms/baselines.

| Name | Description | Value |
|---|---|---|
| lr | learning rate across all modules | 0.0005 |
| optimizer | type of optimizer | RMSProp[1] |
| optim $\alpha$ | RMSProp param | 0.99 |
| optim $\epsilon$ | RMSProp param | $1e-5$ |
| target update interval | copy live params to target params every _ episodes | 200 |
| bs | batch size (# of episodes per batch) | 32 |
| grad clip | reduce global norm of gradients beyond this value | 10 |
| $|D|$ | maximum size of replay buffer (in episodes) | 5000 |
| $\gamma$ | discount factor | 0.99 |
| starting $\epsilon$ | starting value for exploration rate annealing | 1.0 |
| ending $\epsilon$ | ending value for exploration rate annealing | 0.05 |
| $\epsilon$ anneal time | number of steps to anneal exploration rate over | $2e6^{\dagger}/5e5^{\star}$ |
| $h^a$ | hidden dimensions for attention layers | 128 |
| $h^m$ | hidden dimensions for mixing network | 32 |
| # attention heads | Number of attention heads | 4 |
| nonlinearity | type of nonlinearity (outside of mixing net) | ReLU |
| $\lambda$ | Weighting between standard QMIX loss and REFIL loss | 0.5 |
| $\lambda^{\mathrm{AQL}}$ | Entropy loss weight for Amortized $Q$-Learning (AQL) | 0.01 |
| $N_p$ | Number of action proposals for AQL | 32 |
| $N_t$ | Number of timesteps before computing new allocations | $5^{\dagger}/3^{\star}$ |
| starting $\epsilon^p$ | starting value for proposal net sampling exploration annealing | 1.0 |
| ending $\epsilon^p$ | ending value for proposal net sampling exploration annealing | 0.05 |
| $\epsilon^p$ anneal time | number of steps to anneal proposal net sampling exploration over | $3e6^{\dagger}/2e6^{\star}$ |
| starting $\epsilon^r$ | starting value for random allocation exploration rate annealing | 1.0 |
| ending $\epsilon^r$ | ending value for random allocation exploration rate annealing | 0.0 |
| $\epsilon^r$ anneal time | number of steps to anneal random allocation exploration rate over | $7.5e5^{\dagger}/5e5^{\star}$ |

[1]: Tieleman and Hinton [30]
$\dagger$: SAVETHECITY, $\star$: STARCRAFT

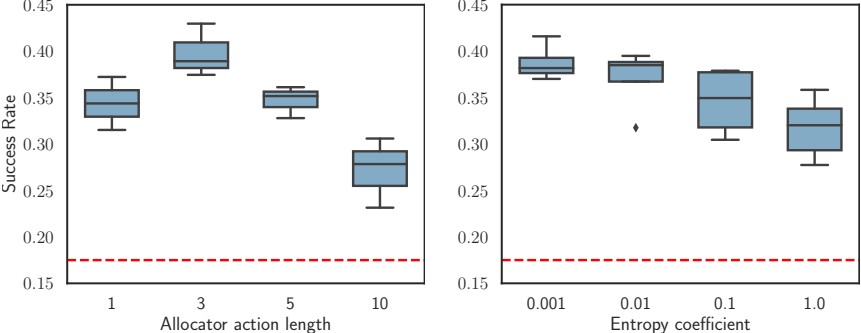

Figure 9: STARCRAFT II (S&Z disadvantage) performance with varying hyperparameter values. Left: varying allocation action lengths. Right: entropy coefficient used in AQL [31]. Red dashed line is performance of best-performing baseline.