# OpenReview forum: "ALMA: Hierarchical Learning for Composite Multi-Agent Tasks"
_NeurIPS.cc/2022/Conference — NeurIPS 2022 Accept_

### Official Review · Reviewer_JvaG · 2022-06-23

**Rating:** 7
**Confidence:** 3
**Soundness:** 3 good
**Presentation:** 4 excellent
**Contribution:** 3 good

**Summary:**

This paper presents a method of hierarchical multi-agent reinforcement learning, specifically for the case where there is a high-level centralized allocator that assigns low-level subtasks to different agents. It assumes that the environment POMDP has some predefined subtasks, from which we can learn low-level policies, and additionally learn the high-level allocator policy simultaneously. The main challenges include handling variable state and action spaces, as well as handling a very large action space for the allocator policy which are addressed via attention models (state) and pointer networks (action). The paper runs experiments on SaveTheCity and StarCraft, two challenging multi-agent environments that involve different number of agents in different configurations. Finally the paper speculates that this formulation may have applications in warehouse robotics.

**Questions:**

L83: "We note that work in multi-agent task allocation often assumes the execution of tasks to be trivial"
L95: "however, they assume the pre-existence of low-level action policies"

Several points were made regarding the difference in ALMA being that it learns low-level action policies as well. But, it seems that we are assuming the POMDP itself has predefined subtasks with its own rewards. To me, this assumption makes learning the low-level action policies trivial, since we can just run any RL algorithm on the subtasks. Is my understanding correct?

**Limitations:**

.

**Strengths And Weaknesses:**

**Strengths**

- The paper is well-executed, in the sense that it identifies practical challenges with the allocator-actor framework and addresses them in very sensible ways. For example, dealing with large action spaces using Amortized Q-Learning and proposal networks, using pointer networks and attention models.

- The experiments are done on challenging environments, and were tweaked to be even slightly more challenging (introducing more enemies in StarCraft), which is impressive. The results are convincing too.

- The writing is exceptionally clear. It gives enough clarification on background techniques and leaves enough room to describe ALMA in detail. Of particular note are the wonderful figures (Figs 2, 3), which are not just helpful/informative, but are truly some of the prettiest I've ever seen.

**Weaknesses**

My main reservation about this paper is the motivation of the problem formulation. There are many, many different setups in hierarchical MARL, each with different assumptions on the environment, observability, centralized/decentralized, etc.... While the Related Work section was helpful, I didn't feel that it was comprehensive enough to really convince me of the importance this allocator/actor framework. Doing a better job in this section would prevent this from feeling like 'yet another MARL framework'.

---

> ### Author Response · Authors · 2022-07-31
> **Response to Reviewer JvaG**
>
> We thank the reviewer for their generous comments, as well as their excellent feedback. We agree that the problem formulation can use more specificity and motivation. If this work is accepted, we can use the additional space granted to expand the Related Work to further differentiate our setting from existing ones and justify its importance. To give a quick summary: our setting is the first to integrate learned task allocation and low-level execution in a modern deep RL framework.
>
> In regards to the reviewer’s question, the reviewer is correct in that learning low-level policies is relatively straightforward when given subtask specific rewards; however, we find that some tricks are required in order to get it working effectively. Most importantly, we find that the masking of subtask information detailed in section 4.2 is crucial for good performance. It is also important to note that there are some nonstationarity issues involved in learning the low-level policies, given that the allocator’s policy changes over time (the reverse is true as well). It is therefore necessary to study the interaction between these two components which we do in this paper, as ALMA is trained end-to-end. We invite a more thorough investigation into these issues as future work.

---

> > ### Comment · Reviewer_JvaG · 2022-08-03
> > **Response to response**
> >
> > Thank you for the response. I have read it, as well as the reviews/response from other reviewers.
> >
> > Just to confirm regarding the differentiation of your setting from existing ones: is it correct that ALMA can handle variable number of agents and variable number of subtasks at test time? Which other methods in literature are able to handle this (or come closest to handling this)?

---

> > > ### Author Response · Authors · 2022-08-04
> > > **Differentiation of Setting**
> > >
> > > Thank you for your response. Yes, that statement regarding ALMA is correct (it can handle a variable number of agents and subtasks at test time). The ability to handle variable numbers of agents comes from using REFIL/Attention-QMIX [1] as our base algorithm for learning low-level controllers. COPA [2] builds on this work to present a hierarchical method for the same multi-task setting. Several other works have also been introduced that can handle variable quantities of agents, though the setting is typically that of curriculum learning, rather than multi-task learning [3-6]. In terms of handling variable quantities of subtasks, we find that the intersection of modern deep MARL and task allocation is relatively understudied. The work in [7] does provide the ability to generalize to variable quantities of subtasks by learning a network that outputs parameters for a linear program solver, which then produces an allocation using constraints provided by domain knowledge; however, this work doesn't solve the low-level learning problem.
> > >
> > > In summary, ALMA is the only method, to our knowledge, that integrates task allocation and execution and can handle variable quantities of agents and subtasks. While each of these problems has been addressed independently in the literature, their combination produces unique algorithmic and engineering challenges.
> > >
> > > [1] Iqbal, Shariq, et al. "Randomized Entity-wise Factorization for Multi-Agent Reinforcement Learning." International Conference on Machine Learning. PMLR, 2021.
> > >
> > > [2] Liu, Bo, et al. "Coach-player multi-agent reinforcement learning for dynamic team composition." International Conference on Machine Learning. PMLR, 2021.
> > >
> > > [3] Long, Qian, et al. "Evolutionary Population Curriculum for Scaling Multi-Agent Reinforcement Learning." International Conference on Learning Representations. 2019.
> > >
> > > [4] Baker, Bowen, et al. "Emergent Tool Use From Multi-Agent Autocurricula." International Conference on Learning Representations. 2019.
> > >
> > > [5] Wang, Weixun, et al. "Action Semantics Network: Considering the Effects of Actions in Multiagent Systems." International Conference on Learning Representations. 2019.
> > >
> > > [6] Hu, Siyi, et al. "UPDeT: Universal Multi-agent RL via Policy Decoupling with Transformers." International Conference on Learning Representations. 2020.
> > >
> > > [7] Carion, Nicolas, et al. "A structured prediction approach for generalization in cooperative multi-agent reinforcement learning." Advances in neural information processing systems 32 (2019).

---

### Official Review · Reviewer_tJvo · 2022-07-11

**Rating:** 5
**Confidence:** 3
**Soundness:** 3 good
**Presentation:** 3 good
**Contribution:** 3 good

**Summary:**

The paper presents a method for achieving hierarchical multi-agent learning. The proposal network is trained to assign subtasks to each agent and each agent will be assigned to accomplish its own task. The proposed method is evaluated on the save the city tasks and starcraft II multi-agent micro-management tasks. It shows better performance on the benchmarks.

**Questions:**

Please see the weakness above.

**Limitations:**

I didn't see some potential negative social impact.

**Strengths And Weaknesses:**

Strength:
1. The paper presents an interesting idea of hierarchical multi-agent learning using the pointer network. The task assignment and skill learning in the multi-agent setting is a good approach and can be applied to many real-world tasks.

2. The writing is clear and the formulation is straightforward.

Weakness:
1. My main concern about the paper is lacking baselines. I think in order to show the superior performance of hierarchical multi-agent learning, we need to compare it against the simple centralized training and decentralized execution method. I didn't see the comparison in the Starcraft setting against papers like QMIX, RODE, ROMA. I think will be convincing if the paper compares the proposed algorithm with these methods, in order to show the benefit of hierarchical learning.

2. I am a little confused about the performance in the Starcraft setting since the results in QMIX, RODE and ROMA are much better than the presented results in FIg. 4. I am wondering if there is some difference in the setting, or is it a fair comparison?

3. I am very interested to see some ablation studies, like the entropy coefficient in Eq. 3 and how the frequency of assigning subtasks to the agent will affect the performance. In particular, if the agent doesn't accomplish the assigned task when the high-level controller decides to change the task, what will happen?

4. It's a minor issue but I feel it will make the paper more convincing if we can test more tasks on the Starcraft settings.

---

> ### Author Response · Authors · 2022-07-31
> **Response to Reviewer tJvo**
>
> We appreciate the reviewer noting the usefulness of the ideas presented in our work and for providing the opportunity to clarify a crucial point: The setting that we build on in this paper is that of multi-task cooperative multi-agent reinforcement learning introduced in [1] and also used by [2]. In this setting, we train cooperative RL agents to solve, not only a single task with a fixed set of agents and entities, but a whole set of tasks with varying sets of agents/entities. While in the single task setting, agents can overfit and learn a single fixed strategy that is able to succeed, the multi-task setting requires agents to be adaptable to a wide variety of settings. As such, the multi-task setting presents unique challenges, and furthermore, single-task MARL methods are not always immediately applicable without significant modification.
>
> For multi-task MARL, the models used in each algorithm must be amenable to the varying size of state/observation spaces across tasks (due to varying numbers of agents/entities). This typically consists of replacing MLP models with some type of entity-wise permutation-invariant architecture (e.g. DeepSets, Attention, GNN, etc.); however, it can sometimes require a more careful adaptation. For example, the appendix of [1] describes the adaptation of QMIX to the multi-task setting, which requires careful consideration due to the need to generate a mixing network of varying size via hypernetworks. In summary, comparing to existing MARL methods on the multi-task setting is not as simple as running the original code on the new setting.
>
> Our baselines were carefully chosen to compare to the state-of-the-art without unnecessary experiments distracting from the primary message. In order to show the usefulness of our hierarchical setup, we compare to the state-of-the-art method in multi-task MARL: REFIL [1]. REFIL first augments QMIX with attention based models and introduces an auxiliary objective which promotes generalization across diverse tasks. Furthermore, we also compare to a SOTA hierarchical method built for the multi-task setting: COPA [2]. Of course, it is sometimes possible to modify single task MARL methods to function in the multi-task setting; however, the modified versions are not validated in the literature. Thus, we choose to focus our experiments on those methods which have been designed for and validated in the setting we choose.
>
> Addressing each of the specific baselines requested:
>
> QMIX) We do, in fact, compare to (attention augmented) QMIX in our SaveTheCity environment, as well as REFIL in StarCraft (Attention QMIX with an auxiliary objective function which thoroughly outperforms QMIX in multi-task StarCraft settings).
>
> ROMA) In [1] they find that REFIL and QMIX both outperform ROMA by a significant margin, possibly due to the fact that role learning is significantly more challenging in a multi-task setting.
>
> RODE) Due to ROMA’s poor performance in the multi-task setting, we omit a comparison to RODE, since it also attempts to learn roles on a per-agent basis and has not been implemented for or validated in the multi-task setting.
>
> We have performed the requested ablations and have included the results as figures in the appendix of the most recent version. We find that ALMA is fairly robust to the choice of entropy coefficient, likely due to the fact that high entropy in the proposal network can be overcome to some extent by the fact that we’re sampling many proposal actions and choosing the best. In terms of subtask reassignment frequency, we do see a reduction in performance when the frequency is too high or too low. When subtasks change too frequently (at every step), we lose the temporal abstraction that hierarchical methods typically provide. When we only allow subtasks to change infrequently (every 10 steps), we find that the agents are not able to react quickly enough to dynamic changes in the environment.
>
> To answer the question specifically: “if the agent doesn't accomplish the assigned task when the high-level controller decides to change the task, what will happen?” We find that the high-level controller typically learns to avoid this scenario, as it will lead to reduced rewards due to the cost of switching between tasks (e.g. in StarCraft agents have to physically move to the location of the newly assigned subtask).
>
> [1] Iqbal, Shariq, et al. "Randomized Entity-wise Factorization for Multi-Agent Reinforcement Learning." International Conference on Machine Learning. PMLR, 2021.
>
> [2] Liu, Bo, et al. "Coach-player multi-agent reinforcement learning for dynamic team composition." International Conference on Machine Learning. PMLR, 2021.

---

> ### Author Response · Authors · 2022-08-08
> **Update requested**
>
> We humbly request that the reviewer acknowledge our response to their review before the end of the discussion period, as we believe all concerns have been thoroughly addressed.

---

> > ### Comment · Reviewer_tJvo · 2022-08-09
> > **Thanks for the response**
> >
> > My concerns are resolved by the authors. Thanks for their response.

---

### Official Review · Reviewer_w9pc · 2022-07-11

**Rating:** 6
**Confidence:** 5
**Soundness:** 3 good
**Presentation:** 3 good
**Contribution:** 3 good

**Summary:**

This paper proposes a general learning method for solving composite tasks in multi-agent RL settings. They propose to learn a high-level allocator with a low-level  action executor. To alleviate problems such as large action spaces, the authors introduce various techniques to make the learning possible. Finally, the authors applied ALMA to a firefighting problems and the SMAC benchmarks.

**Questions:**

Can you discuss more in depth about when your allocator is better than a method with strong heuristics?


**Limitations:**

We appreciate the author for being upfront. The limitations are 1) the exploration are simple e-greedy; 2) observation masks can be discovered automatically.

**Strengths And Weaknesses:**

Strengths:
1. The paper is clearly written.
2. The problem is interesting. RL has shown great progress as a policy optimization tool. However, in many real-world applications, the structure of the tasks should be considered.
3. The performance is better than baseline on various benchmarks.

weaknesses:
1. The generality of ALMA is not convincing. For different tasks, the task allocator might have different action spaces. In some certain problems, the scheduling problem can be solved by other non-deep methods. The authors only empirically show  that the proposed method is better than others without sufficient discussion.
2. The masking trick seems important. But this is pure inductive bias.
3. The testing environment are on the simple end of the spectrum. It would be nice to have a robot task in this paper.

---

> ### Author Response · Authors · 2022-07-31
> **Response to Reviewer w9pc**
>
> We thank the reviewer for their insightful comments and for noting the importance of leveraging the structure of complex tasks for effective learning.
>
> Regarding the first noted weakness, we do agree with the reviewer that ALMA is not a completely general method. ALMA was designed for a specific set of (prevalent) multi-agent tasks, i.e. composite multi-agent tasks where there are several independent objectives required to complete the global task, and agents must balance their resources such that they can accomplish all subtasks.
>
> We would appreciate some clarification on the comment that the task allocator would have different action spaces in different tasks. This is true in the sense that the allocator’s action space scales with the number of agents and subtasks, but this is not a weakness unique to ALMA. In fact, ALMA is designed in a modular fashion (e.g. with pointer nets and attention modules) so that the same network can be transferred across different tasks, which we demonstrate in this paper.
>
> In some cases it is true that scheduling problems can be solved by non-deep methods; however, it is not clear to us why this is a mark against ALMA’s generality. The fact that ALMA can be applied in these settings, in addition to those where conventional methods (e.g. MIP solvers) are intractable or difficult to specify, instead supports ALMA’s generality.
>
> We request that the reviewer clarify what manner of discussion is referred to in the last sentence of the first weakness bullet. We are happy to include such discussion in the final report, but are unsure what specifically is currently lacking.
>
> We agree with the reviewer that the observation masking trick leverages the structure of the problems we are interested in via an inductive bias; however, we do not view this as a weakness, per say. The reviewer themself notes that “in many real-world applications, the structure of the tasks should be considered.” This trick can be leveraged in many real world applications where the independence assumption (even roughly) holds true.
>
> While the testing environments are relatively simple compared to real-world settings, we must emphasize that they are significantly more complex than most existing benchmarks in cooperative multi-agent reinforcement learning. Specifically, in regards to the StarCraft settings, we build on the multi-task settings presented by [1] and used by [2]. These settings are already more challenging than the standard StarCraft tasks, as they require generalization across a wide variety of combinations of unit types, rather than allowing the agent to overfit its strategies to a specific set. Furthermore, we introduce the challenge of multiple armies attacking a central base that must be protected, forcing agents to balance priorities and coordinate at a high level in addition to learning low-level combat strategies.
>
> Finally, to answer the question: the allocation heuristics are designed to assign agents to subtasks in a greedy fashion, optimizing for a ranked order of prioritized metrics (moving to the next metric to break ties). While intuitive to design, these heuristics, even if crafted by human domain experts, are unable to capture the complexity of the task. For example, in StarCraft the “matching” heuristic attempts to match agents to enemy armies such that the unit type compositions of each army match up with the enemies. This ensures that each battle is a “fair fight.” However, ALMA is able to learn a superior strategy which takes advantage of the fact that all enemies are not attacking at once, so the agents can outnumber currently attacking armies in order to gain an advantage while keeping an eye on other armies which may begin attacking. Our attempts to re-capture this behavior via hand-designed heuristic were ineffective, as it was extremely difficult to hand-code the agents to balance priorities gracefully (defending the currently attacking armies vs. sufficiently monitoring dormant armies). The heuristics used in our experiments were the strongest that we (domain experts) could discover (though we acknowledge that this does not constitute proper scientific evidence). We invite future work to validate the difficulty of generating useful hand-designed heuristics in the settings presented in this paper, and we anticipate future work utilizing ALMA-like approaches to beat established heuristics in even more challenging domains.
>
> [1] Iqbal, Shariq, et al. "Randomized Entity-wise Factorization for Multi-Agent Reinforcement Learning." International Conference on Machine Learning. PMLR, 2021.
>
> [2] Liu, Bo, et al. "Coach-player multi-agent reinforcement learning for dynamic team composition." International Conference on Machine Learning. PMLR, 2021.

---

> > ### Comment · Reviewer_w9pc · 2022-08-08
> > **Response to rebuttal**
> >
> > Well received. I'll keep my score.

---

### Meta-Review · Area_Chair_UKS4 · 2022-08-26

**Recommendation:** Accept
**Confidence:** Certain

**Metareview:**

The paper addresses multi-agent coordination in hierarchical task scenario by simultaneously learning coordination and execution policies.

In summary, the reviewers found, and I concur, that the paper would be a welcome contribution to the NeurIPS community. The reviews found the paper clearly written, addressing an important problem, with good results. The rebuttal added new ablations and addressed the most critical reviewers comments. The final version should discuss methods the limitations.

**Award:**

No

---

### Decision · Program_Chairs · 2022-09-14

Accept